# Learning sparse codes from compressed representations with biologically plausible local wiring constraints

**Kion Fallah**[*‡], **Adam A. Willats**[*‡], **Ninghao Liu**[§], **Christopher J. Rozell**[‡]
[‡]Georgia Institute of Technology, Atlanta, GA, 30332 USA
[§]Texas A&M University, College Station, TX, 77843 USA
kion@gatech.edu, awillats3@gatech.edu,
nhliu43@tamu.edu, crozell@gatech.edu

## Abstract

Sparse coding is an important method for unsupervised learning of task-independent features in theoretical neuroscience models of neural coding. While a number of algorithms exist to learn these representations from the statistics of a dataset, they largely ignore the information bottlenecks present in fiber pathways connecting cortical areas. For example, the visual pathway has many fewer neurons transmitting visual information to cortex than the number of photoreceptors. Both empirical and analytic results have recently shown that sparse representations can be learned effectively after performing dimensionality reduction with randomized linear operators, producing latent coefficients that preserve information. Unfortunately, current proposals for sparse coding in the compressed space require a centralized compression process (i.e., dense random matrix) that is biologically unrealistic due to local wiring constraints observed in neural circuits. The main contribution of this paper is to leverage recent results on structured random matrices to propose a theoretical neuroscience model of randomized projections for communication between cortical areas that is consistent with the local wiring constraints observed in neuroanatomy. We show analytically and empirically that unsupervised learning of sparse representations can be performed in the compressed space despite significant local wiring constraints in compression matrices of varying forms (corresponding to different local wiring patterns). Our analysis verifies that even with significant local wiring constraints, the learned representations remain qualitatively similar, have similar quantitative performance in both training and generalization error, and are consistent across many measures with measured macaque V1 receptive fields.

## 1 Introduction

Sensory nervous systems have long been championed for their ability to learn effective representations of natural scene statistics [66]. In fact, this ability has been one of the broad motivations behind work in artificial neural networks underlying significant recent advances in machine learning systems. While it is less understood than the supervised learning frameworks common in machine learning, *unsupervised learning* of task-independent representations is especially important in models of biological sensory systems where supervisory signals are less prevalent. In particular, learning

---

[*]equal contribution
Code available at: https://github.com/siplab-gt/localized-sparse-coding.

a dictionary that admits a sparse latent representation of data (called sparse coding) [49, 50] has become one important theoretical neuroscience model of unsupervised (i.e., self-supervised or auto-encoding [33]) learning in sensory systems. Sparse coding models have successfully explained response properties in sensory cortical areas [60, 73, 71, 72, 38, 67, 12] with biologically plausible implementations [60, 63, 73], as well as playing an important role in machine learning algorithms [1, 25, 46, 55, 45, 9, 57, 58].

While a number of detailed models have been proposed to learn such representations from natural scene statistics in a sparse coding framework (with varying levels of biological plausibility) [49, 73, 60], these models of sensory cortices largely ignore the fact that cortical regions are generally connected by very limited fiber projections. Specifically, there are often information bottlenecks where fewer fibers connect cortical areas than the number of neurons encoding the representation in each area [65], meaning that plausible learning algorithms must account for the fact that the sensory data has undergone some type of compression. For example, despite the original formulation of sparse coding for images nominally constituting a model of neural coding in primary visual cortex [49, 51], the pathway carrying that information from retina to cortex (via the lateral geniculate nucleus of the thalamus) has already undergone a significant reduction in the number of neurons carrying visual information transduced by the retinal photoreceptors [68].

Recently, a theoretical model of these information bottlenecks has been proposed based on randomized linear dimensionality reduction (sometimes called compressed sensing [3]) [35, 16, 32]. In this approach, the sparse coding model is learned in the compressed space, producing latent coefficients that capture the same information as if learning had been performed on the raw data. While this approach is counterintuitive because the learned dictionary has randomized structure when inspected, the key insight is that the receptive fields estimated from this model still resemble primary visual cortex (V1) simple cell receptive fields [35, 16]. Randomized dimensionality reduction has appeal as a neural modeling framework for several reasons [5, 26], including the fact that exact wiring patterns between areas are unlikely to be genetically defined [70] and fiber projections can be specified in this framework with coarse chemical gradients to guide randomized fiber growth during development.

Unfortunately, the current proposals for sparse coding in the compressed space [35, 16] are biologically implausible because the compression operator used is a dense random matrix that violates local wiring constraints observed in many neural pathways. For example, in the early visual pathway, a dense compression matrix corresponds to each retinal ganglion cell (RGC) receiving input from every photoreceptor despite neuroanatomical evidence [68] that RGCs receive inputs from a localized region in visual space. It is currently unknown if a randomized linear dimensionality reduction model for fiber projections in neural systems is feasible under biologically plausible local wiring constraints.

The main contribution of this paper is to leverage recent results on structured random matrices to propose a theoretical neuroscience model of randomized projections for communication between cortical areas that is consistent with the local wiring constraints observed in neuroanatomy. We show that unsupervised learning of sparse representations can be performed in the compressed space despite significant local wiring constraints in compression matrices of varying forms (corresponding to different local wiring patterns). Specifically, we provide both an analytic guarantee on the ability to infer unique sparse representations in this model as well as empirical studies on natural images. Our analysis verifies that even with significant local wiring constraints, the learned representations remain qualitatively similar, have similar quantitative performance in both training and generalization error, and are consistent across many measures with measured macaque V1 receptive fields. Taken together, these results constitute one of the few applications of compressed sensing theory to a specific neural coding model [26], showing that biologically plausible randomized encodings can play a significant role in learning structured representations that capture complex natural image statistics.

## 2 Background and Related Work

### 2.1 Sparse coding

In conventional sparse coding (SC), unlabeled training data is used to learn a representation of the statistics of a dataset through a combination of inference (i.e., estimating the latent variables for a data point in a fixed representation) and learning (i.e., adapting the representation). Concretely, assume that a vector $x \in \mathbb{R}^N$ is k-sparse in an $N \times P$ dictionary matrix $\Psi$ such that $x = \Psi a$ where

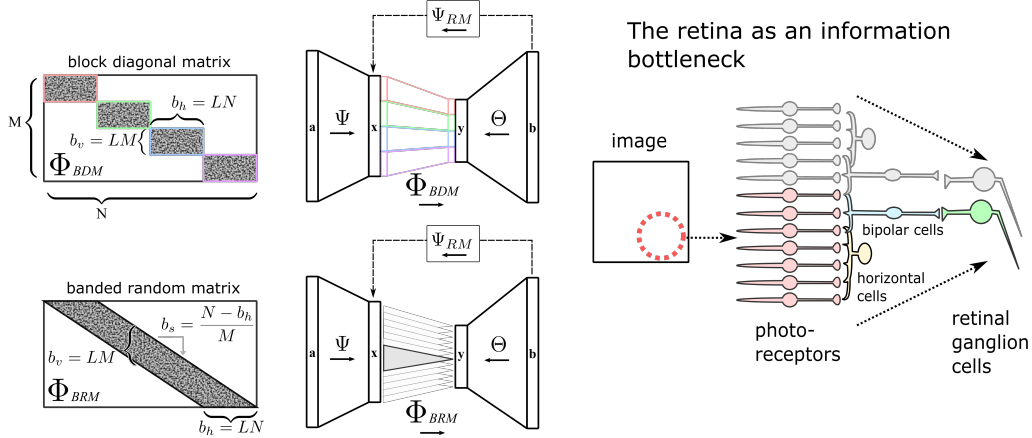

Figure 1: Models of sparse coding on compressed data with with local wiring constraints. We consider two models captured in $M \times N$ random matrices with localization factor $L$: block diagonal matrices (BDM) and banded random matrices (BRM). Sparse coding is performed on compressed data to learn the dictionary $\Theta$ and the recovered matrix $\Psi_{RM}$ is captured by correlating the coefficients $b$ with the inputs (akin to receptive field estimation). These models reflect the information bottleneck of the early visual pathway, where retinal ganglion cells receive input from a localized set of photoreceptors (adapted from [6]).

$a$ has at most $k$ nonzero entries.[2] We follow a common and simple iterative approach to sparse coding [49, 50] using $\ell_1$ regularization that defines an energy function:

$$E(x, \Psi, a) = \|x - \Psi a\|_2^2 + \lambda \|a\|_1. \tag{1}$$

Given a fixed dictionary $\Psi$, inference for a given datapoint $x$ proceeds by minimizing this energy function with respect to $a$ to get the coefficient estimate $\widehat{a}$ and data estimate $\widehat{x} = \Psi \widehat{a}$. Given inferred coefficients for a batch of datapoints, learning is performed by taking a gradient descent step on this energy function with respect to $\Psi$. While SC constitutes a non-convex joint optimization, in practice this approach often converges to reasonable solutions in many natural signal modalities and recent theoretical work has begun to provide recovery guarantees [30, 36, 28, 56, 32, 27]

## 2.2 Randomized linear dimensionality reduction

While there are many approaches to data compression, we concentrate here on randomized linear dimensionality reduction approaches (often appearing in the literature as compressed sensing or compressive sensing [21, 3, 62, 10, 11]) due to the simplicity of biophysical implementation and strong analytic guarantees. This is an established precedent, where previous work with sparse measurement matrices [53] and frame theory [39, 40] have been applied to the recovery of sparse representations of a signal. In these cases, instead of observing $\mathbf{x}$ directly we get the output $\mathbf{y} \in \mathbb{R}^M$ after applying a randomized compression matrix $\Phi$ of size $M \times N$: $\mathbf{y} = \Phi\Psi\mathbf{a} + \mathbf{w}$ where $\mathbf{w}$ represents Gaussian noise. To recover the sparse representation of the original signal, we solve a similar $\ell_1$ minimization:

$$\widehat{\mathbf{a}}(\mathbf{y}) = \arg\min_{\mathbf{a}} \|\mathbf{y} - \Phi\Psi\mathbf{a}\|_2^2 + \lambda\|\mathbf{a}\|_1. \tag{2}$$

Despite the challenging nature of the above problem, analytic guarantees can be made on the recovery. For example, when the matrix $\Phi\Psi$ has properties guaranteeing preservation of information in $k$-sparse signals (e.g., near isometries for signal families, etc.), strong asymptotic recovery guarantees can be made [10]. While it is difficult to deterministically specify or verify matrices that satisfy the necessary conditions for recovery, one can typically prove that $\Phi$ matrices generated randomly from many independent distributions (e.g., Gaussian, Bernoulli) work with high probability for any dictionary $\Psi$ when $M = O(k \log N)$ [59]. These extremely favorable scaling laws and universality properties (i.e., working with any $\Psi$) are due to such independent dense random matrices having

many degrees of freedom and global data aggregation (i.e., every element of **x** is used to form each element of **y**). In cases where the matrix $\Phi$ is structured, such recovery guarantees often require slightly more measurements (i.e., polylog in $N$) and they are no longer universal because the performance changes depending on $\Psi$ [41, 31, 2, 64]. Of special interest for this paper are recent results detailing the properties of random Block Diagonal Matrices (BDM) [52, 23] and Banded Random Matrices (BRM) [13], which are structured sparse matrices requiring only local data aggregation for compression. In the context of sparse coding, this means that the codes inferred for signals compressed with certain matrices will almost surely be equivalent to codes inferred for the uncompressed signals. Recent research has proposed a dynamical systems model for the visual pathway based on these types of random linear projections, but not in the context of learned sensory representations [5].

## 2.3 Dictionary learning from compressed data

In standard compressed sensing formulations, the matrices $\Phi$ and $\Psi$ are considered known. When the dictionary $\Psi$ is unknown, recovering it exactly from compressed measurements is in general an ill-posed problem known as blind compressed sensing [29]. In the case of interest for this paper, the primary concern is the recovery of latent variables $a$ for use in inference tasks and the matrices $\Psi$ and $\Phi$ are both unknown to the downstream cortical area receiving the compressed data.

Previous work [35, 16] proposed that the sparse representations be learned by performing dictionary learning directly on the compressed data (see Figure 1). Specifically, the energy function is given by

$$E(y, \Theta, \mathbf{b}) = \|\mathbf{y} - \Theta\mathbf{b}\|_2^2 + \lambda\|\mathbf{b}\|_1, \tag{3}$$

where $\Theta = \Phi\Psi$. In this approach, dictionary learning directly on **y** is performed according to the alternating minimization procedure outlined earlier by first inferring **b** and then taking gradient descent steps with respect to $\Theta$. Recent analytic results [32] have also guaranteed that the optimal representation of the data can be unique (up to a permutation and a scaling).

While SC in the data space produces dictionary elements that visually resemble V1 receptive fields [49, 50], $\Theta$ appears to have random structure after learning because it is learned in the compressed space. While this initially may appear to conflict with observed biology, the key insight of previous work [35, 16] is that electrophysiology experiments do not measure the synaptic wiring patterns (corresponding to the dictionary elements) but rather measures receptive fields corresponding the the stimulus that most activates a neuron. Using a randomized representation (encoded in synaptic weights) does not preclude structured receptive fields as observed in biology. Indeed, by cross correlating the coefficients **b** to the original inputs **x** (akin to electrophysiologists finding the receptive field of a cell; see Figure 1), previous work shows that it is possible to recover the dictionary $\Psi$ with familiar Gabor wavelet structure and use it for reconstructing the data [35, 16]. Specifically, given access to a subset of data where the uncompressed signals are available, a "reconstruction matrix" (RM) is recovered using $\Psi_{RM} = C_{sr}C_{rr}^{-1}$, where $C_{sr}$ is the cross-correlation matrix $\langle \mathbf{x}\mathbf{b}^T \rangle$ and $C_{rr}$ is $\langle \mathbf{b}\mathbf{b}^T \rangle$. Here $\langle \rangle$ represents the empirical mean with respect to given set of data. As a check to recover the sparse representation of original signal, we can estimate a signal reconstruction $\widehat{\mathbf{x}} = \Psi_{RM}\widehat{\mathbf{b}}$ by solving the original optimization with $\Psi_{RM}$ in place of the dictionary:

$$\widehat{\mathbf{b}} = \arg\min_{\mathbf{b}} \|\mathbf{y} - \Phi\Psi_{RM}\mathbf{b}\|_2^2 + \lambda_t\|\mathbf{b}\|_1. \tag{4}$$

# 3 Learning from compressed data with local wiring

## 3.1 Biologically plausible local wiring constraints in random matrices

In this paper we propose a sparse coding paradigm where learning is performed after compression with a random matrix having biologically plausible local wiring constraints. Wiring minimization has been proposed as constraint faced generally by neural systems throughout development [14, 15]. Given the prevalence of the visual system as inspiration for sparse coding models, we will focus here on the localized wiring patterns observed in the retina as the basis for our theoretical models.

The retina transforms a high-dimensional visual stimulus to a much lower-dimensional neural representation at the optic nerve, with roughly 150 million photoreceptors mapping down to only 1.5 million optic nerve fibers [4]. After photoreceptor transduction, a variety of cell types (bipolar,

amacrine and horizontal cells) process inputs to drive the retinal ganglion cells (RGC) that are the retinal outputs [47]. The simplified schematic in 1 demonstrates that the retina accomplishes this information compression primarily through the pooling of information from nearby photoreceptors [20]. Spatially, RGCs respond to a finite region in the visual field due to the localized wiring in the mapping from the photoreceptors. When examining how visual space is represented at the optic nerve, it can be seen that collectively representations in RGCs cover the full visual space, with each RGC partially overlapping with the representations of its neighbors [20]. In summary, retinal anatomy generally shows a spatially localized pooling of inputs from photoreceptors to neural representation at the level of RGCs. (See supplement, section 6.1 for further discussion of the statistics of retinal connectivity.)

Given what is known about retinal anatomy, we will use two different structured random matrices as theoretical models of compression with biologically plausible local wiring. The first is a block diagonal matrix (BDM), composed of independently randomized sub-blocks along the matrix [52, 23]. The BDM represents a group of RGCs receiving input from a group of photoreceptors, capturing local wiring in a structure that admits strong analysis but potentially exaggerating the partitioning of RGCs into subgroups. The second is a banded random matrix (BRM), composed of independently randomized rows with a compact support that shifts (with overlap) on each row [13]. The BRM represents a model where each subsequent RGC receives input from a limited set of photoreceptors that has partial overlap with the photoreceptors driving neighboring RGCs.

Specifically, each compression matrix is $M \times N$ with $M < N$ and will be characterized by a localization parameter $L \in (0, 1]$ that describes the fraction of each row that has non-zero entries. Each row of a matrix will have $b_h = LN$ non-zero entries. In a BDM, the matrix is composed of $(1/L)$ blocks along the diagonal, consisting of $b_v = LM$ non-zero entries in each column. In a BRM, each row has the non-zero support set shifted to the right by $b_s \in [1, b_h]$ indices relative to the previous row. Up to nominal edge effects, these shifts are constructed so that the last row ends in the lower right of the matrix (or as close as possible), implying that $Mb_s = (N - b_h)$. Note that the $b_s$ parameter is not needed for the BDM. For both matrix types, the number of non-zeros in the matrix is given by $Mb_h = LMN$ and we take the limit of $L = 1$ to be a fully dense matrix. Non-zero entries are drawn from an i.i.d Gaussian distribution of $\mathcal{N}(0, 1/M)$, and this is easily generalized to other sub-Gaussian distributions or normalizations. Note that this model aggregates measurements from across the image (with each measurement formed from a local region), but this is fundamentally different from the common practice of learning a dictionary on individual patches.

Images must be reshaped into a data vector $\mathbf{x}$ in a way that is consistent with the local wiring observed in the retina. For this application, we "unwrap" the 2D image so that nearby neighboring pixels have nearby indices in the 1D $\mathbf{x}$. While there are many possible methods for performing this unwrapping, we use a space-filling curve (the psuedo-Hilbert curve) through the image as the index in $\mathbf{x}$ [37]. As a result, local structure in the compression matrix Phi corresponds approximately to localized regions of 2D image space, much like the spatially localized receptive fields of RGCs. (See supplement section 6.2 for supporting explanation.)

### 3.2 Analytic recovery guarantee

Using recent results on the properties of structured random matrices and recovery conditions for sparse dictionary learning, we can make the following analytic guarantee for when equivalent representations are inferred in the compressed space with a compression process having either of the local wiring structures (BDM or BRM) described above. While the subsequent empirical results will not be limited to specific sparsity dictionaries, for simplicity of exposition, the unified analytic result below is specific to the case where the sparsity dictionary $\Psi$ is an orthonormal Fourier basis (i.e., a standard DFT matrix). This is a reasonable candidate case given the empirical success of JPEG compression, and these results have some extensions to other sparsity bases (as discussed below).

**Theorem 1** *Let $\Phi$ be constructed as either a BDM or BRM $M \times N$ matrix with localization parameter $L$ as described above and fixed. Let $\Psi$ be a Fourier orthobasis and generate a dataset $y_i = \Phi \Psi a_i$ with $a_i$ chosen randomly to be $k$-sparse with $k > 1$ (i.e., uniformly choose the support set of $k$ nonzeros in $b_i$ followed by choosing coefficient values uniformly from $(0, 1]$) and $i = 1, \ldots, D$ for $D = (k + 1)\binom{N}{k}$. If*

$$M \geq Ck \log^2 k \log^2 N \qquad (5)$$

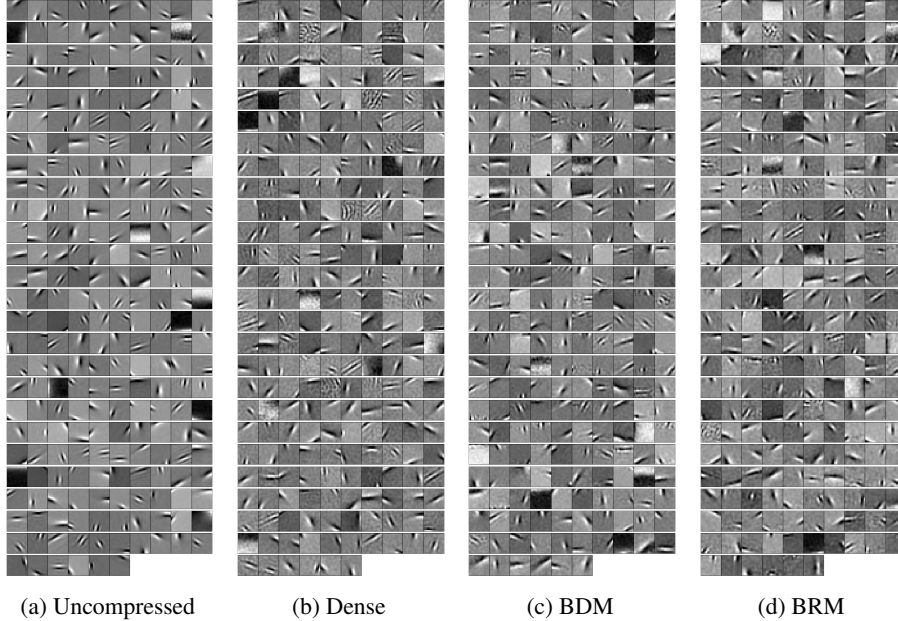

| (a) Uncompressed | (b) Dense | (c) BDM | (d) BRM |

Figure 2: Recovered dictionaries ($\Psi_{RM}$) learned on natural images. (a) Learned on uncompressed patches (conventional SC). (b) Learned after compression with a dense random matrix ($L = 1$). (c) Learned after compression with a block diagonal matrix (BDM) having $L = 1/16$. (d) Learned after compression with a banded random matrix (BRM) having $L = 1/16$. Recovered dictionaries are qualitatively similar despite learning after compression under significant local wiring constraints.

*for a constant $C$, then with high probability (tending to 1 in the limit of the problem size), any k-sparse encoding of the dataset $\{y_i\}$ is equivalent up to an arbitrary permutation and scaling of the columns of $\Phi\Psi$ and the coefficients $\{b_i\}$.*

The proof of Theorem 1 is in the supplement, section 6.3. A few comments and observations about this result are warranted. First, note that the above theorem gives conditions for which the ideal coefficients $b$ for the compressed data are equivalent (up to permutation) to the ideal coefficients $a$ for the uncompressed data. This implies that the system will learn the same representations in the sense that their measured receptive fields will also be equivalent (again, up to permutation). While there is no guarantee that realistic numerical algorithms will find these ideal solutions, our empirical results show that they can learn representations that are very close to those learned in the uncompressed case despite having some conditions of the theorem (e.g., orthogonal bases) not strictly met.

Second, note that the statement of the theorem doesn't explicitly depend on the degree of localization $L$ or the matrix type. While the recovery guarantee doesn't depend on $L$, it is possible that the embedding quality (reflected in the isometry constant $\delta$ or scaling constants) could vary with $L$. While we have unified the result for simple exposition, the result is not tight for BRMs [13] and polylog factors could be reduced. Third, while we have presented the results above to keep the exposition as simple as possible, the theorem above could be generalized to characterize the uniqueness and stablilty of the recovery with respect to measurement and modeling error [27].

Finally, given the structured nature of these random matrices, performance will depend on the sparsity basis. In particular, as discussed [52, 23], matrices with localized measurements require a diversity condition that prevents all energy in the image from concentrating in localized blocks. For BDM, Theorem 1 can be extended trivially to any sparsity basis, with performance depending on the incoherence of the basis with the canonical basis (Fourier has optimal incoherence). While general BRM results have not been derived [13], it is likely that similar general results could be shown.

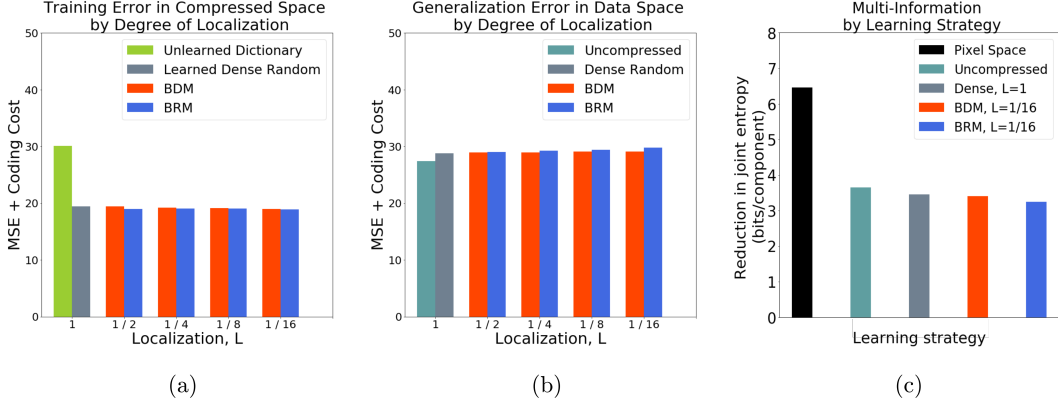

Figure 3: Coding performance of learned representations. (a) Sparse coding objective (2) for compressed patches using the compressed dictionaries. Unlearned dictionary is equivalent to random initialization before learning, demonstrating that learning improves coding even for randomly compressed data. (b) Generalization error computed with sparse coding objective (1) on patches from hold-out data-set with reconstructed dictionaries $\Psi_{RM}$. (c) Redundancy reduction, as measured through joint entropy, per learning strategy. Uncompressed dictionary is learned with uncompressed patches (i.e. conventional sparse coding). Despite learning with significant compression and wiring constraints, training and generalization is comparable to conventional sparse coding.

## 4   Experimental results

To test the proposed model, we conduct a number of learning experiments using whitened natural image patches compressed with the BDMs and BRMs with varying degrees of localization ($L$) and compression ratio $M = 0.5N$ (other compression ratios did not qualitativly change results, shown in the supplementary materials). Specifically, in our experiments we used $80,000$ $16 \times 16$ patches extracted from 8 whitened natural images for training. This was broken into batches of size 100, iterated over 150 epochs with decaying step-size on the learning gradient step. To infer coefficients we used $\lambda = 5e - 2$, chosen experimentally as a value that produces stable learning convergence. These hyper-parameters were held constant across all experiments. We kept 10% of the training data-set uncompressed for correlation in recovering $\Psi_{RM}$.

Figure 2 shows the results from learning on natural images with different compression matrices and degrees of localization. This includes uncompressed learning (i.e., conventional SC), and $L = 1$, corresponding to the unconstrained compression performed in [35, 16]. The plotted dictionaries for the compressed methods correspond to $\Psi_{RM}$ recovered from correlating uncompressed patches as in [35, 16]. Qualitatively, the recovered dictionaries appear visually similar to conventional SC even with compression and significant levels of localization, demonstrating that biologically plausible local wiring constraints can still yield receptive fields similar to those observed in V1 simple cells.

For quantitative evaluation on the effect of localization, we examine the ability of the learned representations to minimize the coding objectives. This objective is computed in both the compressed space, as well as generalization performance in the (uncompressed) data space. First, Figure 3(a) shows the value of the coding objective in the compressed space (2) evaluated on the training data-set. Figure 3(b) uses the corresponding recovered dictionaries, $\Psi_{RM}$, to compute the data space objective (1) on a validation data-set for different degrees of localization. The validation data-set was built from $20,000$ patches extracted from 2 images not used in training. Note that although these objectives are similar, the compressed space has reduced dimensionality, meaning fewer pixels need to be estimated, thus leading to lower error. Notice that in both measures, there is little change in the objective due to compression or due to the degree of localization. Note also that we plot the training error of an unlearned dictionary, demonstrating that there is significant value to learning the representation even when the data has been subjected to randomized compression. In addition, Figure 3(c) follows previous methods [8, 24] in depicting the redundancy reduction in all learned dictionaries (with small additional reductions when learned in the compressed space, and further small reductions when wiring is localized).

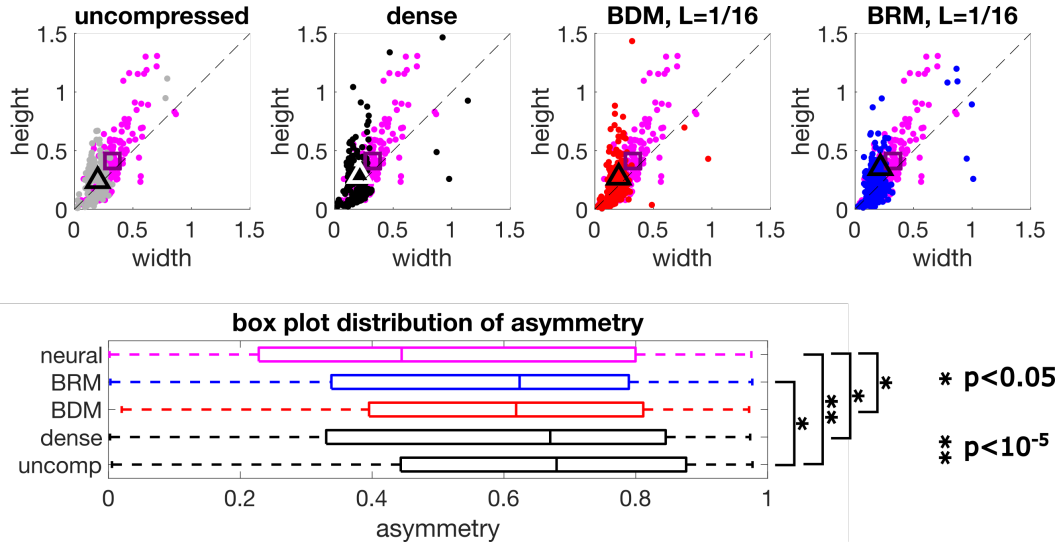

Figure 4: Comparing receptive field characteristics between learned models and macaque V1 measurements [61]. (a, top) This row shows distribution of height and width of gabor wavelet receptive field fits for macaque V1 (magenta) compared to a variety of learned models, including uncompressed, dense matrix ($L = 1$), block diagonal ($L = 1/16$) and banded ($L = 1/16$). Distribution centroids are marked with triangles for dictionary learning and squares for centroid of V1 neural data. Learning in the compressed space results in receptive field shapes which are qualitatively similar to observed neural data, even for high degrees of localization (L=1/16). (b, below) Box-plot representation of distribution of asymmetry wherein the boundaries of the box represent 25th and 75th percentile of the data as well as the median. Receptive field asymmetry for BRM dictionaries is significantly lower than that of uncompressed matrices ($P < 0.05$), and asymmetry measured from neural data is significantly lower than the uncompressed, dense and BDM models ($P < 0.05$).

Finally, we evaluate the recovered representation by comparing the statistical distribution of learned receptive field shapes to those recovered from electrophysiology experiments in macaque V1 [61]. Following [60] and similar to [61], we fit each recovered dictionary element in $\Psi_{RM}$ with a 2D Gabor function $A \exp\left[-\left(\frac{u'}{\sqrt{2}\sigma_{u'}}\right)^2 - \left(\frac{v'}{\sqrt{2}\sigma_{v'}}\right)^2\right]\cos(2\pi f u' + \phi)$, where $u'$ and $v'$ are transformed image coordinates, widths of the Gaussian envelopes are given by $\sigma_{u'}$ and $\sigma_{v'}$, and $f$ is the spatial frequency [22]. Note that envelope width is given by $W = \sigma_{u'}f$ and envelope height (referred to as envelope length in [60]) is given by $H = \sigma_{v'}f$, measuring the extent of the element relative to the number of spatial frequency cycles it spans. Asymmetry is calculated via the normalized difference of the integral on each side of the $v'$ midline as in [60]. Eccentricity, defined as $|\log(H/W))|$, measures the aspect ratio of the Gabor envelope. Eccentricity is 0 for circularly symmetric receptive fields and larger for more elongated elliptical receptive fields.

Figure 4 quantitatively compares the statistics of the learned dictionary elements. Data from [61], (magenta points in figure 4 (a)) show that neural receptive fields cover a wide range of shapes. Previously, it was shown that sparse coding produces a better match than Independent Component Analysis (ICA) [7] to observed cortical receptive fields in macaque V1 [61]. Here we see that a similar distribution of receptive fields characteristics (broadly matching experimentally observed receptive fields) are recapitulated during dictionary learning with a variety of compression strategies, including the significant local wiring constraints of our biologically plausible model.

While the distribution of receptive field widths and heights recovered using the approaches presented here are generally smaller than those recovered from neural data (see also supplemental figure 9), a precise match to receptive field size was not a primary goal of this work. A better match to observed neural data might be achievable with a more overcomplete dictionary or by incorporating a nonlinear output stage.

Due to the importance of asymmetry, used to distinguish learned representations in past research [61], we performed Kruskal-Wallis analysis of variance on the asymmetry to assess whether the learned representations shown in figure 4 (a) all come from the same distribution. We found evidence to reject this null hypothesis with $P < 10^{-5}$. However, with further multiple comparisons tests, we saw that receptive field asymmetry for BRM was significantly lower than receptive fields recovered with uncompressed matrices (Figure 4(b), $P < 0.05$). Moreover, receptive fields recovered from V1 electrophysiology data were less asymmetric (Figure 4(b)), than those recovered with each method other than BRM (each $P < 0.05$). Taken together, these results demonstrate that the proposed model learns representations that fit experimental data at least as well as the conventional SC model, with the BRM model potentially fitting experimental data better than learning with uncompressed matrices.

## 5   Conclusions

We have shown for the first time that unsupervised learning of sparse representations can be performed on data compressed using random matrices with biologically plausible local wiring constraints. Using a variety of structure random matrix models, we provide an analytic recovery guarantee as well as empirical characterization showing that the proposed model learns a representation that is at least as good of a fit to visual cortex electrophysiology data as the conventional sparse coding model. These results lead us to conclude that randomized compression with biologically plausible local wiring constraints is a compelling model for information bottlenecks in fiber projections, and the statistics of natural images are highly amenable to unsupervised learning in the proposed model.

## Broader Impact

This paper presents a theoretical neuroscience model that aims to help improve our understanding of neural coding in biological sensory systems. While this may eventually have secondary effects as inspiration for machine learning systems, the current result is general and theoretical enough that we do not foresee particular applications or negative societal consequences. The images used in the experiments of this paper were the same as those used in previous work on sparse coding [50].

## Acknowledgments and Disclosure of Funding

This work is partially supported by NSF CAREER award CCF-1350954.

## Footnotes

[2]Many results in the literature generalize gracefully in the case that data is only approximately $k$-sparse.

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
