[Supplementary Material]

# 6 Supplemental material

## 6.1 Statistics of retinal wiring

One key goal of this work is to introduce an approach to sparse coding which respects the spatially localized connectivity present in the early visual pathway (from images projected onto the retina, to representations in the retinal ganglion cells). In the visual system, each RGC is connected to a subset of photoreceptors which gather light from a contiguous, roughly circular region of an image termed its receptive field. It has been shown that the receptive fields of individual RGCs partially overlap with those of neighboring RGCs. In summary the connectivity of the visual system is sparse and spatially localized. Our intention is to capture the *essence* of local wiring in the retina, as such support of each row of $\Phi$ are restricted to a small range. However neither the block/band size nor the degree of overlap are intended to be a direct fit to levels of localization and overlap in the biological retina. Instead we focus on characterizing a range of levels of localization to better understand the theoretical and empirical implications of this approach.

The properties of the retina are highly non-uniform in space, with a central fovea containing a high density of cones ($\sim 50 kcells/mm^2$) and an outer periphery of relatively higher density of rods (at peak, $\sim 150 kcells/mm^2$) [18, 42]. The density of RGCs peaks at an intermediate distance ($\sim 30 kcells/mm^2$) and decreases further out [17, 44]. From this, we can infer that the ratio of photoreceptors to RGCs varies between close to 2:1 or 1:1 near the fovea, up to 100s of rods:1 RGC in the periphery. Including the pooling of photoreceptors introduced by horizontal and amacrine cells each RGC may pool up anywhere from 1 to 1000 nearby photoreceptors [19] depending on cell-type and location in the retina. In summary, retinal anatomy generally shows a spatially localized pooling of inputs from photoreceptors to neural representation at the level of RGCs.

## 6.2 Spatially-localized image pre-processing

As part of the sparse coding approach we transform a 2D image into a 1D vector $\mathbf{x}$. In order to emulate the local connectivity constraints of the retina, we would like neighboring pixels in 2D image-space to map to neighboring pixels in 1D vector-space. To accomplish this we borrow the Hilbert curve (figure 5), because of its simple construction and locality-preserving property. This step involves generating a space-filling curve which traces a path through the pixels of the image, then assigning indices in the 1D vector according to their position along the curve [37].

It has been shown that using Hilbert curves, nearby indices in 1D dimension are mapped to nearby pixels in 2D. While the converse is not always guaranteed (that neighboring pixels in 2D must map to neighboring elements of their 1D vector) it has been shown that the Hilbert curve approach maintains locality better than several alternatives [48]. As such, it has been applied to various applications from image compression [69, 43] to storing geographic coordinates in contiguous segments of a 1D memory address register [54].

Because the vector $\mathbf{x}$ uses a spatially contiguous representation of the image, and because of the local structure present in the compression matrix $\Phi$, local regions of image space also correspond to localized segments of $\mathbf{y}$. This is further illustrated in figure 6 where we show smoothly varying 2D images map to qualitatively smooth 1D vectors.

## 6.3 Proof of Theorem 1

Draw a random matrix according to the construction specified in section 3.2 and fix it. Under the conditions of the theorem, we first establish that for either random matrix construction, the combined matrix $\Phi\Psi$ satisfies a Restricted Isometry Property (RIP)

$$(1-\delta)||\mathbf{a}||_2^2 \leq ||\Phi\Psi a||_2^2 \leq (1+\delta)||\mathbf{a}||_2^2 \tag{6}$$

for all $a$ that are $2k$-sparse (i.e., $\|a\|_0 \leq 2k$) with $0 \leq \delta < 1$.

First we consider BDM matrices. Theorem 1 in [23] establishes that BDM matrices satisfy the RIP with isometry constant $0 < \delta < 1$ for $M \geq Ck \log^2 k \log^2 N$ with $C$ that depends on $\delta$ and with probability greater than $O(1 - N^{-\log N \log^2 k})$. Note that Theorem 1 in [23] depends on the incoherence of the sparsity dictionary with the canonical basis (denoted $\widetilde{\mu}$ in Theorem 1 of [23]), which is established as $\widetilde{\mu} = 1$ for the Fourier basis in [23]. We also observe that this result is

Figure 5: Procedure for preserving locality in models of sparse coding with wiring constraints. While images are represented as values in 2D space, the sparse coding approach presented here operates on 1D vectors. In order to map pixel locations to vector indices, a Hilbert curve is drawn through the image. Positions along the curve correspond to indices in the vector which can be thought of as unwrapping the curve into a line. By combining this unwrapping with compressed sensing matrices, representations in this method maintain locality from 2D image space, to the compressed representation.

modifiable in a straightforward way for other sparsity bases with known incoherence. Finally, we also note that the failure probably goes to zero as $N \to \infty$.

Next we consider BRM martices, with the same discarding of edge effects in the BRM matrix as done in [13]. Theorem 2 in [13] establishes that BRM matrices satisfy the RIP with isometry constant $0 < \delta < 1$ for $M \geq Ck \left( \log(N/k) + 1 \right)$ with $C$ that depends on $\delta$ and with probability greater than $1 - 2e^{-CM\delta^2}$. Examining these conditions, we see that

$$k \left( \log(N/k) + 1 \right) = k \left( \log(N) - \log(k) + 1 \right)$$
$$< \tau k \left( \log_2(N) \right) < \widetilde{\tau} k \left( \log^2(N) \log^2(k) \right),$$

where the second line follows as long as $k > 1$, with $\tau$ and $\widetilde{\tau}$ constants due to a change of log base. The final quantity establishes that the conditions of the present theorem ensure that the conditions of Theorem 2 in [13] hold. While we have presented one unified result for both matrix types, note that this result is not tight for the BRM matrices and a much more aggressive compression (i.e., lower $M$) is permissible, but at the expense of generality to other type of sparsity bases. Finally, we also note that the failure probably goes to zero as $M \to \infty$, which is necessary as $N \to \infty$.

When RIP holds, for any pair of $k$-sparse vectors $a_1$ and $a_2$, $\Phi\Psi(a_1 - a_2) = 0$ implies that $a_1 = a_2$. This can be restated as

$$\Phi\Psi a_1 = \Phi\Psi a_2 \implies a_1 = a_2,$$

which is known as the *spark condition* that characterizes the linear dependence structure of the columns of $\Phi\Psi$. Given the spark condition satisfied for $\Phi\Psi$, Corollary 2 in [32] establishes that with probability one, for $D = (k+1)\binom{N}{k}$ datapoints drawn randomly (i.e., uniformly choose the support set of nonzeros in $a$ followed by choosing coefficient values uniformly from $(0,1]$), any $k$-sparse encoding of the dataset is equivalent up to an arbitrary permutation and scaling of the columns of $\Phi\Psi$ and coefficients $b_i$. ∎

Figure 6: Illustration of preserved image features after unwrapping. (a) smoothly-varying pixel values for a 16x16 image (b) The Hilbert curve method involves assigning an order to pixels based on the curve's path shown in black. (c) Using the curve's ordering, pixels in 2D space are arranged or unraveled into a 256x1 vector. Pixel values are seen to relate to their spatial neighbors in a smoothly-varying fashion qualitatively similar to the original 2D image. (d) An alternative approach would be to simply stack the columns to reshape the image. The equivalent path through 2D space is again visualized in black. (e) As a result of the discontinuities between columns, the image flattened with this approach exhibits periodic jumps in pixel intensity not present in the original 2D image.

## 6.4 Result of changing compression

Additional tests were run to determine the effect of the compression ratio on the generalization error. This error was computed with the sparse coding objective in the data space (as in (1)), with patches not used in training. With a constant localization $L = 1/4$ for the BDM and BRM matrices, we tested compression ratios of $1/2$, $3/8$, and $1/4$. The results can be seen in Figure 7, where we see a small increase in generalization error with more compression. Note that as the compression ratio increases, the measurement function has fewer parameters to support gathering information for learning and more of the training data is discarded. It can also be seen that the generalization error is only slightly higher for localized matrices at each compression ratio.

## 6.5 Gabor feature quantification

To understand the characteristics of receptive fields estimated through our technique, we quantified width, height, asymmetry and eccentricity of the receptive fields (see experimental results, figure 4). Figure 8 compares the two dimensional distribution of asymmetry and eccentricity between receptive fields learned through the methods presented here, to those fit to neural visual system data. The distributions cover a similar range across methods.

In figure 9 we show the one-dimensional distribution of each of these features and compare across methods. Width and height occupy a similar range across all conditions except the macaque neural data which has larger Gabor envelopes.

In testing for significant differences in Gabor features across methods, we performed a multiple comparisons test. This extends the Kruskal-Wallis analysis shown in the main text, for which the null hypothesis is that all groups of data come from the same underlying distribution. From this analysis,

Figure 7: Coding performance of learned representations with varying degrees of compression and localization held constant to $L = 1/4$. Tested on $20,000$ validation patches not used in training.

Figure 8: Comparing receptive field asymmetry and eccentricity between learned models and macaque V1 measurements [61]. Asymmetry is calculated from the normalized difference in intensity above and below the midline of the wavelet. Eccentricity is measured as $abs(log_{10}(H/W))$. Distribution centroids are marked with triangles for dictionary learning and squares for centroid of V1 neural data. Learning in the compressed space results in receptive field shapes which are qualitatively similar to observed neural data, even for high degrees of localization (L=1/16).

we have evidence to reject the null hypothesis at a significance level $\alpha = 0.05$ with $P = 3.6 * 10^{-6}$. In this multiple comparisons test, we look at pairs of groups (i.e. uncompressed versus BRM) and calculate P-values shown in table 1. For these tests, the null hypothesis is that data from the two groups comes from the same distribution. As such, these P-values represent an upper bound on the probability of falsely identifying significant differences between groups [34].

|  | dense | BDM | BRM | neural |
|---|---|---|---|---|
| uncompressed | 0.298 | 0.226 | **0.040** | **9.11e-06** |
| dense |  | 0.999 | 0.906 | **0.025** |
| BDM | " |  | 0.951 | **0.038** |
| BRM | " | " |  | 0.221 |

Table 1: P-values from comparing asymmetry across pairs of conditions through Kruskal-Wallis analysis of variance. BDM, BRM are shown for localization $L = 1/16$. P-values less than a significance level of $\alpha = 0.05$ are highlighted in bold. The null hypothesis that the medians of all groups are equal can be rejected with $P = 3.6 * 10^{-6}$

Figure 9: Comparing 1D distribution of receptive field characteristics between learned models and macaque V1 measurements [61]. Box plots shows distributions of Gabor features across conditions. Median, 25th, and 75th percentile shown as box boundaries. BDM, BRM are shown for localization $L = 1/16$