[Reviews · NeurIPS 2020]

Review 1

Summary and Contributions: The authors bring forward a well-known shortcoming in typical sparse coding models, which is that the operations performed by the retina are largely attributed to some sort of whitening transform that does not perform the dimensionality reduction that we can see when looking at the physical constraints imposed by the optic nerve (e.g. ration of photoreceptors and ganglion cells). Previous works have addressed this to show that sparse coding on compressed inputs using dense random compression matrices still recovers the original codes. They extend this previous work to show that it also works with sparse compression matrices. I have not seen this particular result before, although my exposure to this specific extension of sparse coding is admittedly limited.

Strengths: The paper brings together a few different works in an interesting way, namely: (1) sparse coding effectively recovers (up to some conditions) the true generating sparse codes under noise constraints [22], even if the data has first been compressed [27, 29, 15] and (2) compression can be performed by a sparse random matrix [40, 19, 12] instead of a dense one. The authors theorem of stating that the banded or group matrices work as well is interesting.

Weaknesses: There are a few caveats for their quantitative measure. Ref [46] argued that their "hard thresholding" constraint (basically using matching pursuit instead of basis pursuit denoising) improved the biological match. However, Rehn & Sommer changed the thresholding method and increased overcompleteness compared to their control sparse coding model. It was later argued by Olshausen that the overcompleteness likely had a larger impact than the thresholding type. I bring this up because there are a lot of factors that play into how well receptive fields match up by this measure, including how they perform whitening/preprocessing, overcompleteness, and how they perform the sparse inference. In the abstract and conclusion the authors are careful to say that they have similar performance to regular SC, which is fair as long as they keep preprocessing, overcompleteness (P/N), and the sparse inference method fixed for all models tested. My point here is that the metric they use can be swayed pretty far in one direction or another based on parameters & methods that they do not mention anywhere in the main text, so I'd feel more comfortable if they were explicit about that. In Bethge et al. (2005) and Eichhorn et al. (2009, Plos CB) it is argued that the widely perceived goal of sparse coding with orientation selective filters, namely to reduce redundancy in the natural image representation, is only partially reached by the resulting codes. The authors develop a suite of quantitative metrics to compare sparse coding models which would be also interesting to apply here. The authors should discuss the point that all their models including the ones with localized structure fail to account for the correct width/height distribution, especially for the width.

Correctness: As far as I can tell, the paper is correct.

Clarity: Overall the paper is sufficiently clear, but the authors could put in some extra effort to accommodate non-experts. Lines 186-191 are confusing. Please explain this in more detail. It would also be helpful if the motivation given in lines 154ff and the changes they make to the paradigm were more clearly explained in the Intro.

Relation to Prior Work: The authors should relate their work to PCA whitening + dimensionality reduction that is always done as a pre-processing step by Hyvarinen and colleagues when doing ICA. Hyvarinen et al. do dimensionality reduction via removal of low-variance PCA dimensions instead of a matrix product and I think Hyvarinen's y dimensionality is larger than what is considered here, but it is dimensionality reduction none the less. They also should consider work from Cottrell & colleagues which combines PCA dimensionality reduction with ICA to learn filters that look like RGC & V1 filters. Olshausen & colleagues also do implicit dimension reduction by low-pass filtering their inputs as part of the whitening transform, although this is much farther from their proposal than the first two examples. All of these methods are different in flavor from the compressed sensing work which is probably why they were overlooked. However, especially for the Hyvarinen & Cottrell works the result is quite similar: Dimensionality reduction + whitening followed by sparse coding produces good-looking RFs. What's more, the linked Cottrell model also includes locality constraints in the sparse PCA "retina" step to make it more biologically plausible. I think the contribution by the authors is still novel / valuable because of the analytical theorem and connection to the compressed sensing literature.

Reproducibility: Yes

Additional Feedback: {202} I think that equation should be y_i = Phi Psi a_i if I am following the diagram in figure 1 correctly. As far as I can tell, they are generating a dataset of images, x, from ground-truth sparse codes, a. Then they compress the image with Phi to get y. Then they do sparse coding y = Theta b (eq 3) to estimate the new sparse codes. The proof is trying to say that a = b up to some simple transforms. {219} typo "could be affected vary with" If we assume center/surround (difference of gaussians) receptive fields then we get whitening, but how does that match up with the randomness of the compression matrix? I guess I don't understand what the structure of the compression matrix tells us about the RFs of RGCs (e.g. {164-172}). The authors could comment a bit more in how far BRM/BDMs are good models for RGCs.


Review 2

Summary and Contributions: The basic question addressed in this paper is whether it is possible to achieve sparse, lower-dimensional representations of signals (as achieved e.g. by multiplying by dense random matrix in compressed sensing) using an information processing approach that respects local wiring constraints that are seen in neurobiology. This is motivated by unsupervised learning of task-independent representations of sensory data. Drawing on results on structured random matrices, the answer is yes. This finding is then interpreted in the context of mammalian sensory processing, e.g. as exemplified by receptive fields of macaque V1. The authors have thoughtfully engaged with the reviews, and I believe the paper will be stronger after revision.

Strengths: * Asking the question of wiring constraints in the sparse coding paradigm is a good question to ask to move closer to biological plausibility * Mathematical approach drawing on DCT basis is a nice way to introduce structure * Macaque experiments seem to be well-done

Weaknesses: * Some more details on the statistical properties of anatomical wiring in sensory cortex would have been helpful to better indicate how sparse the measurement matrices must be * Greater discussion of related literature on sparse measurement matrices in compressed sensing would have clarified novelty. Likewise some discussion of frames in harmonic analysis may have been helpful to the reader. * Figures are hard to read due to small size

Correctness: To this reviewer, the paper appears correct.

Clarity: Please make figures larger so as to make more legible. Writing is well-done.

Relation to Prior Work: Some greater discussion of how this work is related to work on compressed sensing with sparse sensing matrices (just one example is Parvaresh, Vikalo, Misra, and Hassibi, "Recovering Sparse Signals Using Sparse Measurement Matrices in Compressed DNA Microarrays," 2008, but there is a whole literature) would be appreciated. Likewise, some more discussion of frame theory, going back to the simplest example in the field of the so-called Mercendes-Benz frame which is a submatrix of the DFT matrix would help place the work better in the literature.

Reproducibility: Yes

Additional Feedback: Broader Impacts may want to include details about the 8 whitened natural images that were used for training in the experiments.


Review 3

Summary and Contributions: The authors explore different compression matrices incorporating realistic biological constraints and demonstrate that sparse representations can be learned from these compressed projections. The results in this paper demonstrate that sparse conding models are not really challenged by anatomically realistic wiring constraints and therefore remain plausible models of information transmission in the brain.

Strengths: The authors proposal is well-backed by both theoretical and experimental results. The analytic result is sound and the empirical validation is extensive enough with qualitative as well as quantitave results. This work continues a line of research exploring the validity of sparse coding models of information transmission in the brain, which has already made important contributions in the past (as noted by the authors), but was significantly challenged by anatomical evidence. This work successfully demonstrates that sparse coding models remain plausible models of information transmission, and paves the way for these models to keep making important contributions to our understanding of efficient information coding both in biological and artificial neural networks.

Weaknesses: The work is solid and continues an interesting line of research exploring the feasibiity of structured random matrices as wiring models of information compression in the brain. I see no obvious weaknesses in this submission.

Correctness: All the claims seem correct and the methodology used in the paper is sound and convincing.

Clarity: The paper is very clearly written.

Relation to Prior Work: The relation to prior work is clearly discussed in the paper.

Reproducibility: Yes

Additional Feedback:

[Author Response · NeurIPS 2020]

We are grateful to the reviewers for their substantive and constructive feedback. We are grateful that there is consensus that this manuscript **(R2)** "brings together a few different works in an interesting way" to **(R3)** "move closer to biological plausibility" with sparse coding models in a way that is **(R4)** "well-backed by both theoretical and experimental results" and **(R4)** "paves the way for these models to keep making important contributions to our understanding of efficient information coding both in biological and artificial neural networks." We propose revisions and are additional experiments (detailed below) to address reviewer comments, and believe the resulting manuscript will be much stronger.

While we appreciate that **(R4)** viewed the manuscript favorably, we're also confident that there are opportunities for improvement. Unfortunately, without listing any weaknesses, we are unclear what changes **(R4)** could envision that would improve the impact of the work.

**1: Connections to related research.**

- **(R2)** We agree that a number of modeling factors affect the dictionary structure and we will improve the manuscript with an explicit discussion of these points. As noted, the goal of our study is not to produce the highest fidelity with biological receptive fields among all possible models, but rather to focus on the role of localized wiring constraints in randomized compression for the sparse coding model (which itself did not perfectly match the width/height distribution). To be explicit, we used the same hyper-parameters across all experiments. While our focus is specifically on the localized wiring, we also note that the prior work introducing learning in the compressed space (with dense wiring) did not report quantitative aspects of coding performance or fits with biological data. Similarly, we will further contextualize our results by including a discussions of the relationship to other papers involving dimensionality reduction and sparse coding (including the works by Hyvarinen and Cottrell).

- **(R3)** We agree that we can improve the context of the manuscript by highlighting the relationship to other results on structured sparse measurement matrices as well as frame theory from the harmonic analysis literature. This change will be included in the revision.

**2: Additional analysis.**

- **(R2)** We agree that the redundancy reduction metrics would be interesting and valuable to add to the manuscript (as another panel in figure 3). In fact, we have already calculated those metrics on the recovered dictionaries and we see qualitatively the same relationship as with the other metrics. Specifically, this analysis shows that multi-information is significantly reduced from the pixel space in all cases, with similar levels of redundancy reduction with all learned dictionaries (with small additional reductions when learned in the compressed space, and further small reductions when wiring is localized).

- **(R3)** We agree that additional analysis based on the retinal neuroanatomy literature will help place our modeling choices in better context. Specifically, we will include a detailed discussion of photoreceptor density (ranging from 50k-150k cells/mm$^2$), retinal ganglion cell density (peaking at 30k cells / mm$^2$), and degree of photoreceptor innervation of individual retinal ganglion cells (ranging from 1-1000). This analysis will further demonstrate that retinal anatomy generally shows a spatially localized pooling of inputs from photoreceptors in a way that is qualitatively captured by the proposed model (with characteristics that quantitatively reflect many of the basic properties of the biology).

Figure 1: *New analysis of redundancy reduction showing similar decreases in redundancy for all learned dictionaries.*

**3: Writing clarity.**

- **(R2)** We agree that the material in lines 186-191 and line 154 is less clear than it should be. We will revise this text to be more clear, including adding text and a graphic to the supplement to explain the reshaping procedure.

- **(R3)** The figures will be revised to increase readability.

[Meta-Review · NeurIPS 2020]

Three knowledgeable referees support acceptance for the contributions, notably for proposing a theoretical neuroscience model of randomized projections for communication between cortical areas, and I also recommend acceptance. The reviewers were happy with the rebuttal and agreed that the paper was well-backed by theoretical results and that the empirical validation was good with a balance of qualitative and quantitative results. The authors have thoughtfully engaged with the reviews, and I believe the paper will be stronger after revision.